# General practitioner workforce sustainability to maximise effective and equitable patient care: a realist review protocol

Sophie Park,[1] Emily Owen-Boukra [iD],[2] Bryan Burford [iD],[3] Tanya Cohen,[2] Claire Duddy [iD],[1] Harry Dunn,[4] Vacha Fadia,[2] Claire Goodman [iD],[5] Cecily Henry,[2] Elizabeth I Lamb,[3] Margaret Ogden,[2] Tim Rapley [iD],[6] Eliot Rees,[2,7] Gillian Vance,[3] Geoff Wong[1]

For numbered affiliations see end of article.

**Correspondence to**
Dr Emily Owen-Boukra;
emily.owen@ucl.ac.uk

## ABSTRACT

**Introduction** There are not enough general practitioners (GPs) in the UK National Health Service. This problem is worse in areas of the country where poverty and underinvestment in health and social care mean patients experience poorer health compared with wealthier regions. Encouraging more doctors to choose and continue in a GP career is a government priority. This review will examine which aspects of the healthcare system affect GP workforce sustainability, how, why and for whom.

**Methods and analysis** A realist review is a theory-driven interpretive approach to evidence synthesis, that brings together qualitative, quantitative, mixed-methods research and grey literature. We will use a realist approach to synthesise data from the available published literature to refine an evidence-based programme theory that will identify the important contextual factors and underlying mechanisms that underpin observed outcomes relating to GP workforce sustainability. Our review will follow Pawson's five iterative stages: (1) finding existing theories, (2) searching for evidence, (3) article selection, (4) data extraction and (5) synthesising evidence and drawing conclusions. We will work closely with key stakeholders and embed patient and public involvement throughout the review process to refine the focus of the review and enhance the impact and relevance of our research.

**Ethics and dissemination** This review does not require formal ethical approval as it draws on secondary data from published articles and grey literature. Findings will be disseminated through multiple channels, including publication in peer-reviewed journals, at national and international conferences, and other digital scholarly communication tools such as video summaries, X and blog posts.

**PROSPERO registration number** CRD42023395583.

## INTRODUCTION

UK general practice has been described as being in crisis.[1] A shortage and exodus[2] of general practitioners (GPs) is an urgent and challenging problem, attracting significant media attention and widespread public

## STRENGTHS AND LIMITATIONS OF THIS STUDY

⇒ This is the first systematic review to use realist methodology to explore which aspects of the healthcare system affect general practitioner workforce sustainability, how, why and for whom.

⇒ We will include a diverse range of evidence, including, empirical articles, conference materials, policy reports and editorials, increasing review depth and breadth.

⇒ We will embed patient and public involvement and stakeholders throughout the design, analysis and report stages of the project, enhancing impact and relevance.

⇒ We will only include documents that are written or translated into English.

⇒ Our context-mechanism-outcome configurations and refined programme theory may be limited by the availability, quality and richness of existing identified literature in this field.

debate. The National Health Service (NHS) Long Term Plan proposed a net increase of 5000 GPs, alongside expectations of work distribution and reallocation across primary care teams.[3] The impact of these changes on the GP workforce remains relatively unexplored. In the UK, general practice provides comprehensive (a patient may present with any type of problem), person-centred (holistic caring and individualised care) and universal (everyone can attend) access to healthcare. The ways in which this care is organised and delivered do tend, therefore, to differ depending on the needs of local communities, available resources and capacity.[4]

It is widely acknowledged that general practice provides 90% of patient care using approximately 8% of the NHS budget,[5] meaning this is a key element of broader debates about a whole-system NHS moral and

financial crisis. Existing research has focused on 'numbers in' 'numbers out' calculations of workforce need, predictions and requirements. GPs' decisions to stay or leave are often positioned as an individual 'choice',[6–8] resilience or failure, with less attention given to the social and organisational milieu in which those decisions are produced. A systems approach recognises the need to acknowledge this inherent dynamic complexity, by studying the interconnected components (eg, the nature of social interactions between patients, colleagues, and external institutions, alignment of personal and policy priorities, and organisational support structures) that work together in an integrated and coordinated way to sustain and enhance the GP workforce.

The figures reflecting shifts in the number of GPs available are well rehearsed. For example, as of February 2023, the number of full-time equivalent GPs has fallen significantly (approximately 2100 fewer fully qualified full-time GPs compared with September 2015),[9] many GPs are approaching traditional retirement ages, and the challenges of recruiting and retaining GPs in rural and coastal areas are well recognised, both nationally and internationally.[10] UK GPs have reported higher levels of emotional distress than GPs in nearly all other countries.[11] In parallel, readily measurable elements of GP work have been reported to increase, with 37.2 million consultations estimated to have been delivered by practices and primary care networks in October 2023, of which 2.9 million were COVID-19 vaccinations.[12] Consultations are reported to have increased in length,[13] alongside changes in form (eg, use of technology). Measures of GP time spent on operational challenges and problems are high[14] and the 'hidden', non-patient-facing work of GPs is increasingly being recognised.[15] It is widely acknowledged that GPs staying in practice has benefits for both patient care[16 17] and practice productivity.[18] For example, experienced GPs work more efficiently, lessening the scale of patient medicalisation, investigation and treatment, and requiring shorter appointment times compared with trainees and other healthcare professionals.[19–21] We can also see that practices with a higher proportion of patients in deprived areas; lower care quality commission ratings and single-handed and rural practices, have fewer GPs overall and more locums[22] emphasising the importance of wider contextual factors in shaping the nature of work and workforce patterns.[23]

A systems approach is well suited as a basis for examining the multitude of factors influencing the GP workforce and how these ultimately shape career decisions. By taking a system and theory-driven realist approach to understanding what works, in which contextual circumstances and to what extent, we aim to make visible the often unsaid or implicit issues that influence working environments, relationships and organisational culture. The findings from our review will offer structural and organisational recommendations to inform sustainable, context-specific ways of future working, which can underpin effective and equitable patient care. For the purposes of our review, we define equitable patient care as the process through which healthcare systems maximise inclusive ways to facilitate access and care delivery for patients. While equality implies delivery of the same for all, equity focuses on ensuring patients have the best opportunity to reach needed outcomes, even if systems must change or adapt processes, to enable this. Effective patient care may vary according to the patient's needs and contextual circumstances but will include the best possible fit for them at that particular point in time.[4]

## METHODS
### Review aim, research objectives and research questions
#### Aim
This review aims to examine which aspects of the healthcare system affect GP workforce sustainability, how, why and for whom.

### Research objectives
1. Develop a programme theory through an evidence synthesis of how the GP workforce can be sustainable and work effectively with others to support effective and equitable patient care.
2. Embed and respond to patient and public involvement (PPI) and stakeholder perspectives throughout the design, analysis and report stages of the project, thus maximising the relevance and utility of review findings.
3. Make recommendations for practice and policy based on the refined programme theory.

### Research questions
What can we learn from the existing literature that will promote GP workforce sustainability, to support effective and equitable patient care? Specifically:
1. Which mechanisms produce intended and/or unintended outcomes in GP workforce sustainability?
2. What are the important contexts which influence whether different mechanisms produce intended and/or unintended outcomes in GP workforce sustainability?
3. Which interventional strategies lead to intended and/or unintended outcomes for GP workforce sustainability?

#### Approach
We will conduct a realist review, synthesising data from the published literature to produce a refined evidence-based programme theory. A programme theory explains how a programme, intervention or process works (eg, how we can develop a more sustainable GP workforce).[24] The programme theory will identify the important contexts (conditions or circumstances) and mechanisms (underlying causal forces) which underpin observed outcomes (intended and unintended) relating to GP workforce sustainability to support the delivery of effective and equitable patient care. A realist review is an interpretive and theory-driven approach to evidence synthesis that brings

together data drawn from quantitative, qualitative and mixed-methods research, as well as the wider grey literature.[25] Using these data and a realist logic of analysis will allow us to examine diverse evidence with a clear focus on understanding factors which support or challenge GP workforce sustainability, how these are shaped by important contexts and the mechanisms that underpin them. Based on this understanding, captured in our final programme theory, we will be able to identify and prioritise important system-level contexts that may be amenable to change. Our review began in January 2023 and will end in December 2025.

Initial scoping searches using EMBASE, MEDLINE and PsycINFO identified a range of qualitative and quantitative primary and secondary evidence focused on GP retention. We have identified four existing reviews focused on GP retention,[7 26–28] and four additional studies from broader scoping searches, including one examining factors associated with part-time working.[29] Existing studies on this topic are often descriptive, identifying a complex array of factors associated with workforce attrition without theoretical analysis. These include individual issues (eg, well-being, burnout and identity), practice (eg, workload, administrative burden and clinical autonomy[26])—and opportunities for relational continuity of care[7 27] and national (eg, revalidation) contexts.[1 30–32] One study found the intrinsic attractions of retirement (eg, increased time for leisure) contributed to GPs leaving practice.[33] Factors shaping GP retention in one UK region[34] were described as 'push' (workload, stress, job dissatisfaction and organisational injustice demotivating factors) and 'pull' (work–life balance).

This review will attend to the complex systems in which GPs work, in order to better understand the factors which, keep, sustain and enable GPs to flourish within their work. A realist approach will allow us to examine a wide range of literature about the system and related social interactions (eg, across the primary and secondary care interface, between patients and clinicians) and to integrate patient and stakeholder experience to inform our approach and produce new ways of thinking about this challenging field. We will identify causal explanations to illuminate the connections between GP work (eg, nature and scope) and GP career intentions. Depending on the available evidence, we aim to examine a range of contexts (eg, location, employment arrangements such as small or large institutions, salaried, partnered, payment by performance), systems of support inside or outside practices and patient deprivation. We will examine the relationships between these contextual factors and a range of reported outcomes. This will help to identify the circumstances in which a sustainable and flourishing GP workforce for the delivery of effective and equitable patient care can be realised.

Our approach will follow Pawson's[25] five iterative stages for realist reviews detailed below. This begins with an initial programme theory developed through scoping searches and preliminary consultation with stakeholders and patients. As the review progresses, we will gather and interpret data from the literature to refine and develop our explanations of important outcomes, drawing on the expertise of our PPI and stakeholder groups. At the end of the project, our final programme theory will consist of evidenced context-mechanism-outcome configurations (CMOCs), presenting realist causal explanations for outcomes relating to the GP workforce sustainability to support effective and equitable patient care.

## Patient and public involvement

We have three PPI co-applicants within our core research team. The team also includes professionals from a range of disciplinary backgrounds including general practice, secondary care, nursing, psychology and sociology. Additionally, we have recruited six PPI contributors from coastal, rural and diverse urban contexts. An equality of opportunity approach will be taken throughout this process, enabling us to respond in an agile way to maximise PPI involvement. For example, helping to refine the programme theory, shape our findings and contribute to both the format and content of our outputs (eg, written word and infographics), and ensuring that proposed solutions are both feasible and acceptable. Finally, our PPI members will play an essential role in developing our dissemination strategy, and the format and content of the outputs of this project. We also have a stakeholder group including professionals representing a variety of GP employment models, paramedic, pharmacy, policy, practice, management and educational backgrounds.

In the development of this proposal, we invited PPI and stakeholders to share their priorities. GPs often work at the interface between human experience and the technicalities of pathology, negotiating boundaries and overlaps between stress, illness and disease.[35] Person-centred care has been recognised as a key element of this process.[36 37] Knowledge of the 'person' can be established over time, building on trust and relational currency between patient and practitioner.[38] One key area of concern in PPI and stakeholder discussions during the development of this review protocol was 'care of other'. GPs spoke about the importance of person-centred care while PPI co-applicants highlighted the importance of looking after practitioners and supporting their well-being. As one PPI contributor explained, if patients are treated as 'a number' within the healthcare system, this is potentially mirrored in their treatment and expectations of healthcare professionals. One key area of GP workforce sustainability in this review, therefore, is the relational elements of GP work, and how these may support or undermine workforce sustainability.

We invited PPI and stakeholders to help produce three case study examples about GP workforce sustainability:

### Case study 1

In the UK, following the introduction of the NHS Long Term Plan, multidisciplinary team working in general practice is increasing.[3] Many practices now employ pharmacists, paramedics, physician associates, physiotherapists

and mental health professionals. In some circumstances, this has enabled GPs to focus on the clinical care of particularly 'complex' patients (eg, with multimorbidity), delegate some home visits[39] and prescribing demands.[40] Yet, some stakeholders report (a) anecdotal increases in workload relating to supervision and support of new staff and (b) increased levels of follow-up (eg, lower thresholds for initiating patient investigations), (c) patients needing to attend multiple appointments to resolve a problem(s) and (d) challenges negotiating access for patients with unselected and comprehensive healthcare problems to see the most appropriate professional, at the most appropriate time. These fast-moving changes to the GP role and patient access to GP care have, therefore, been experienced both as producing opportunities and challenges.

### Case study 2
Relationships between GPs and secondary care colleagues can influence patient care and access to services. For example, if close and rapid communication is established (eg, informal sharing of numbers, or formal routes for email advice and guidance), GPs can progress patient care, minimising referrals and waiting list delays, and maximising timely patient care delivery in the primary care setting. This could have an impact on a GP's sense of agency or autonomy to provide immediate care for patients, but also a sense of working as part of a distributed team, rather than alone. As an example: a specialist nurse managing a paediatric allergy clinic advice service. In one area, this enables access to advice regarding initial investigations and treatment. In another area, this increases recommendations to refer. This means waiting lists could be shortened or lengthened, expediting or delaying assessment of urgent or severe patients, or periods of time requiring additional GP support.

### Case study 3
GPs sometimes develop 'special interests' (eg, supporting patients with mental health problems). This can be positive for practices, patients and GP professional development. Yet, it can also produce challenges and inequalities if not managed well. For example, patients requiring more clinical input, longer consultations and emotional investment being directed to seek help from particular GPs. Without adequate planning, this can result in GPs who support these patients running late, working longer hours (for no extra pay) to complete other routine tasks. Without sufficient resources and support, this can affect GP well-being and potential burnout.

### Step 1: locating existing theories
We have developed an initial programme theory to describe our assumptions about important influences on GP workforce sustainability. This outlines potentially important contexts, active mechanisms and outcomes relating to GP workforce sustainability that we need to consider or build on over the course of the review. Important contexts, for example, might include organisation size, employment type, practice skill mix and nature of patient contact. Mechanisms could comprise continuity, including, for instance, relational, informational, longitudinal and management types of continuity,[41 42] peer dialogue or clinical autonomy. A draft of our initial programme theory is provided in figure 1.

### Engaging PPI and stakeholder perspectives in expanding the initial programme theory: exploring how to create joy
To expand our initial programme theory and shape initial searches, we carefully planned a series of individual and small group discussions with PPI and stakeholder members to open a conversation about 'what creates joy

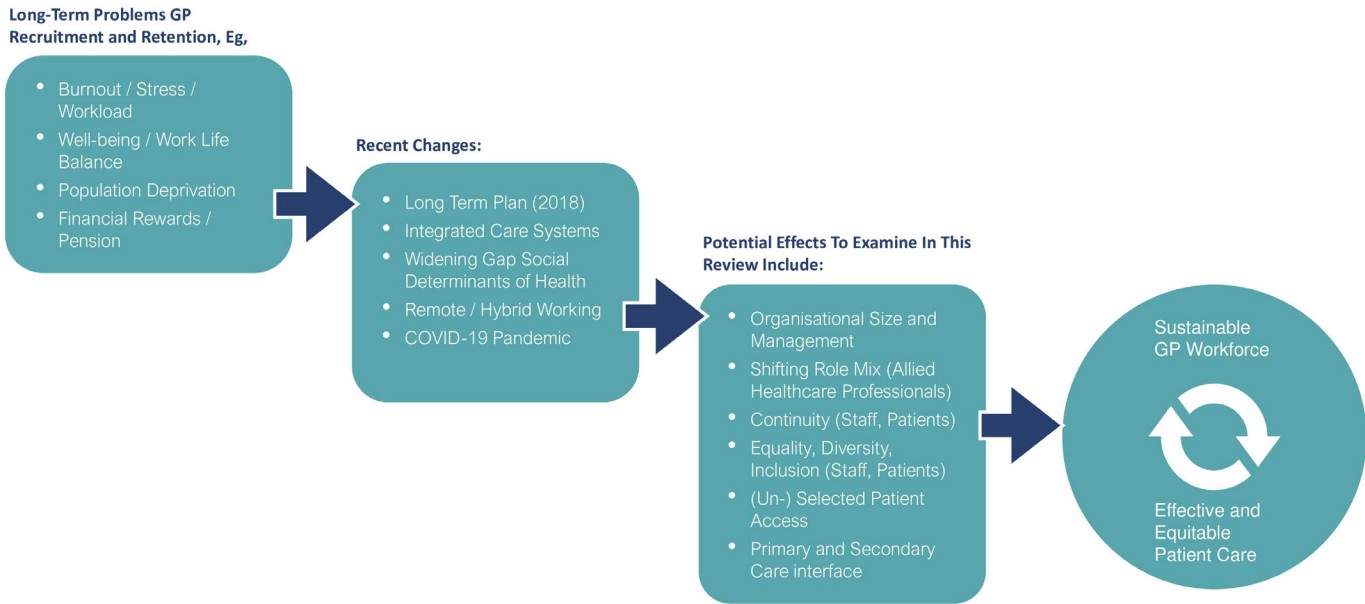

**Figure 1** Initial programme theory—used at the beginning of realist review projects to map initial explanatory theories. GP, general practitioner.

in general practice and enables (or produces conditions for) 'effective and equitable patient care'. We focused on joy (and meaning making) as most of the literature we identified in our initial scoping searches was predominantly negative, highlighting the reasons why GPs leave or consider leaving direct patient care. We recognise that many GPs continue to enjoy and derive pleasure from their work.[43] As such, our aim throughout the PPI and stakeholder discussions was to understand the factors that keep GPs engaged meaningfully in their work. Participants were invited to draw on their own experiences and perspectives to explore this topic. Following introductions, we asked our PPI and stakeholder members to draw, describe or write words which represented their thoughts, feelings or experiences about creating joy in general practice and enabling effective and equitable patient care. These were probed and discussed to build up a map of issues, including system-level factors (eg, patient access, nature of consultation interactions, the interface between team members and other organisations, explicit patient-facing work and other 'hidden' work such as administration tasks). This map helped to modify and expand our initial programme theory, which was then used to shape initial searches and became the starting point for our review. An overview of the key concepts from our PPI and stakeholder discussions can be found in our word cloud (figure 2). A draft of our expanded initial programme theory is provided in figure 3.

### Step 2: searching for evidence

We will conduct searches to assemble a relevant body of literature that contains data we can use to develop and refine the initial programme theory developed in step 1. To help us work efficiently, we will reuse search strategies employed in our scoping review and extend and update these to identify new, relevant material also indexed in additional databases (CINAHL, HMIC, Web of Science—SCI-EXPANDED and SSCI indexes). The main databases we will search include MEDLINE, Embase, PsycINFO, CINAHL, HMIC and Web of Science. These searches will combine free text and subject heading (MeSH) terms for GPs with a range of terms relating to workforce and retention outcomes. The full details of the search developed for MEDLINE are available in online supplemental file 1. We will conduct searches from other data sources including relevant organisational websites (eg, Health Foundation, King's Fund, Royal College of General Practitioners, Department of Health and Social Care) to identify grey literature.

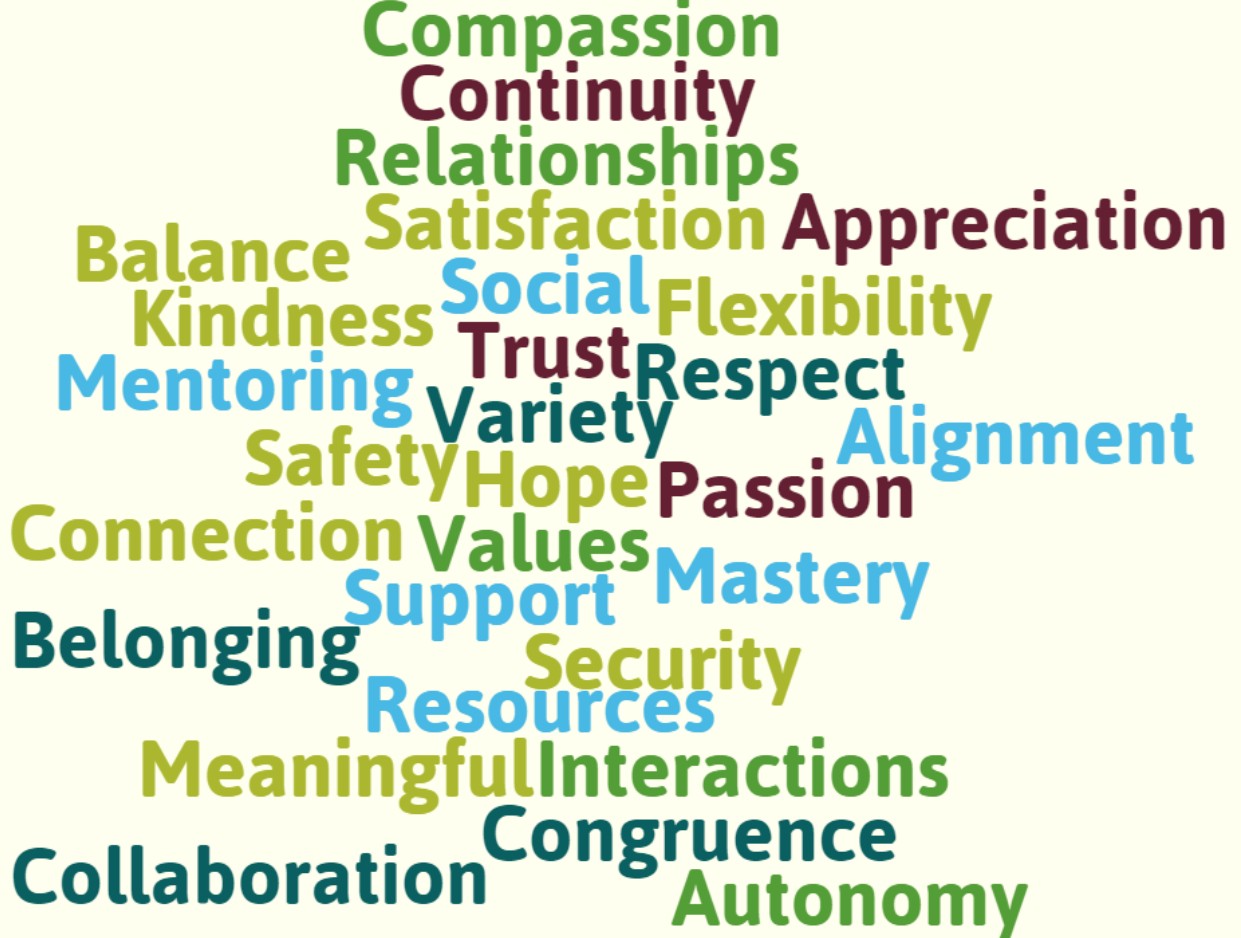

**Figure 2**  Word cloud visualisation.

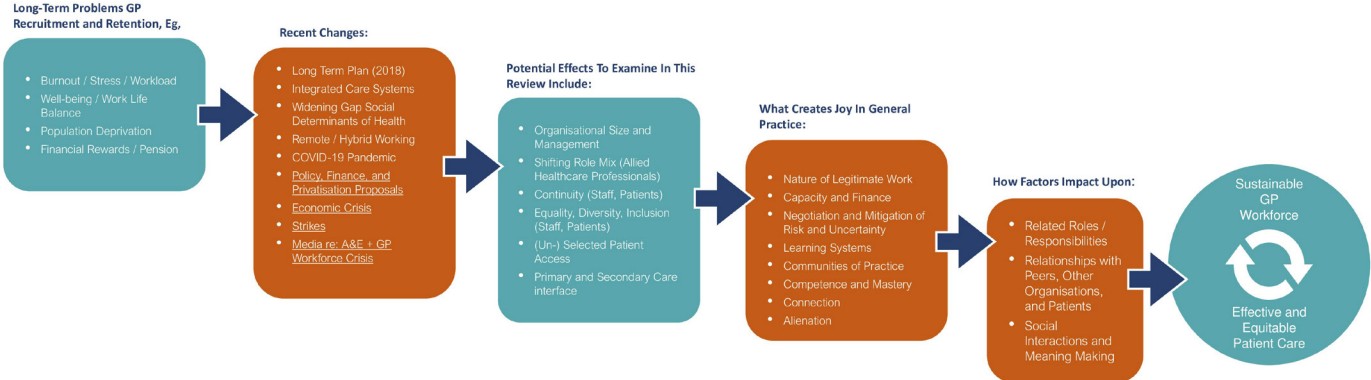

**Figure 3** Expanded initial programme theory after stakeholder and PPI consultation workshops. GP, general practitioner; PPI, patient and public involvement.

We will limit our searches to identify material published in the English language. No limits will be placed on study design, as all relevant evidence (including opinion and commentary) can be used to inform programme theory development. As the review progresses, we will identify additional material via citation searching, our PPI and stakeholder members, and alerting services. Our initial inclusion criteria will include GP as the object of discussion or participant, discussion or study of UK general practice setting (including out-of-hours), include enquiry or discussion of GP workforce and/or GP work (eg, nature, role, change, shift), relationship between GP and work or meaning attributed to work, and documents published within the last 10 years.

Throughout the review, discussions within the project team and with PPI and stakeholders will be ongoing, framing the focus and direction of literature selection, extraction, and analysis, and informing the production of the final programme theory configurations. At each stage, the team will ask 'is this relevant to the programme theory?', focusing on the connection between 'GP workforce sustainability' and 'effective and equitable patient care' to inform decisions about whether to explore and include certain areas of evidence and literature.

If necessary, we will conduct additional searches later in the project, with the aim of identifying data to fill gaps or refine specific aspects of our developing programme theory. Our additional searches will be guided by the input of our PPI and stakeholder groups, as we work with them to prioritise areas of the developing programme theory for exploration. Crucially, we will explore key workforce sustainability concepts (see above) with our PPI and stakeholders, alongside the United Nations definition of sustainable practice which includes the working environment, resources and people interacting within that system.[14]

### Step 3: article selection

We will screen documents for inclusion in the review in a three-step process: screening by title/abstract, following by screening in full text, with the final selection resting on an assessment of each document's relevance (whether it contains data relating to relevant contexts, mechanisms, outcomes or the relationships between these) and rigour (whether the methods used to generate each piece of data are credible and trustworthy).[25] To ensure consistency in this process, a 10% sample of decisions made at each stage will be independently checked by research team members (including CD, ER, BB and EIL). Any discrepancies or disagreements will be discussed with EO-B and the research team, and an agreed resolution documented.

### Step 4: data extraction

Included full-text documents will be uploaded to NVivo (qualitative data analysis software) for coding. Coding will be inductive (codes created to categorise data as reported in the included documents), deductive (some codes will be created in advance, based on the initial programme theory in step 1) and retroductive (codes created based on an interpretation of the data, to infer what hidden causal forces may generate outcomes). Each new piece of relevant data identified will be used to refine and develop our programme theory and as the review progresses, included documents will be rescrutinised to identify relevant data that may have been missed earlier, or as the programme theory developed. As with our screening and selection processes, a 10% sample of included documents will be independently checked by CD, ER, BB and EIL to ensure consistency. Any discrepancies or disagreements will be discussed within the research team and documented.

### Step 5: synthesising the evidence and drawing conclusions

We will use a realist logic of analysis to make sense of the extracted data and develop CMOCs that offer explanations for outcomes relating to GP workforce sustainability and the provision of effective, equitable patient care. We will use interpretive cross-case comparison to understand and explain how and why observed outcomes occur. For example, we will compare documents that describe contexts which are reported to enable GPs to flourish and sustain their working roles, to understand how contexts influence GP workforce sustainability and support effective and equitable patient care.

We will use a proven analysis and synthesis process.[44] In brief, to operationalise our realist analysis, we will ask the

following questions about the included documents and data therein:

► Interpretation of meaning: do the documents provide data that may be interpreted as functioning as context(s), mechanism(s) or outcome(s)?
► Interpretations and judgements about CMOCs: what is the CMOC for the data that has been interpreted as functioning as context, mechanism or outcome?
► Interpretations and judgements about programme theory: how does this CMOC relate to the initial programme theory?

At the end of the process, we will produce a refined programme theory, underpinned by the data extracted from included documents, on which our co-production activities will build for dissemination of review findings.

## ETHICS AND DISSEMINATION
### Dissemination

Our PPI co-applicants and stakeholders will help us to decide on the content, storyboarding and format (eg, websites, leaflets, videos, social media). In addition to the final report, we will produce:

1. Academic outputs; for example, protocol publication. Findings (to be submitted to a high-impact, open-access and peer-reviewed journal), as well as tailored papers to different disciplinary journals.
2. Audience-specific practitioner 'how to' publications which outline practical advice on ways to maximise the sustainability of the GP workforce to support effective and equitable patient care.
3. User-friendly summaries of the review findings tailored to the needs of different audiences including the public and service users.

Our dissemination strategy will build on a participatory approach, embedding PPI co-applicants and stakeholder involvement throughout the development of this research and project timeline, including opportunities for co-authorship and co-presenting. Ongoing engagement with key stakeholders will maximise opportunities to use our established networks, communication channels and links to policymakers and providers. Our approach will be integrative, valuing the different forms of knowledge needed to produce findings capable of informing complex decision-making within policy and practice.

Our audiences are as follows:

1. Policy-makers, health system decision-makers and commissioners will be key to implementing recommendations from our review.
2. Providers and practitioners whom we will brief on our findings.
3. Members of the public and charities—for whom we will tailor press releases, social media posts and engage directly via our PPI and stakeholder networks. Our migrant PPI members will help devise ways to best disseminate our outputs to non-English-speaking communities.

### Ethics

Ethical approval is not required for this review as only secondary data sources will be used.

## DISCUSSION
### Importance of the research

Following an escalation in changes affecting general practice during and after the COVID-19 pandemic, there are national and international debates about what general practice of the future should look like.[45–47] Improving recruitment and retention of GPs should be a high priority in medical workforce planning. Yet, with many GPs reducing their working hours or retiring early, new theoretical perspectives and research are needed to examine how we can support and retain the GPs we have, alongside enabling the recruitment of new GPs. We hope the findings from our review can influence political discussions and shape future patient care, contributing to the scholarship of the field and to a more sustainable general practice. Throughout our research, we will generate new knowledge about the interdependencies between contextual factors, causal mechanisms and outcomes of interest. The findings may be used to inform strategies and interventions intended to support, facilitate and assist the GP workforce in delivering effective and equitable patient care. We will identify critical gaps in knowledge and prioritise the expectations for the scope and nature of GP work.

**Author affiliations**
[1]Nuffield Department of Primary Care Health Sciences, University of Oxford, Oxford, UK
[2]Department of Primary Care and Population Health, University College London, London, UK
[3]School of Medicine, Newcastle University, Newcastle upon Tyne, UK
[4]School of Clinical Medicine, University of Cambridge, Cambridge, UK
[5]Centre for Research in Public Health and Community Care, University of Hertfordshire, Hatfield, UK
[6]Department of Social Work, Education and Community Wellbeing, Northumbria University, Newcastle upon Tyne, UK
[7]School of Medicine, Keele University, Keele, UK

**Contributors** The realist review was conceptualised by SP. SP wrote the first draft of this protocol and it was reviewed and revised by EO-B, BB, TC, CD, HD, VF, CG, CH, EIL, MO, TR, ER, GV and GW. The search strategy was developed by CD in consultation with EO-B and the wider project team. All authors read and approved the final manuscript. SP is the guarantor.

**Funding** This project is funded by the National Institute for Health Research (NIHR) School for Primary Care Research (SPCR593). TR is a deputy theme lead within the NIHR Applied Research Collaboration Northeast and North Cumbria (NIHR200173). Emily Owen-Boukra, Sophie Park and Geoff Wong are members of the NIHR SPCR Evidence Synthesis Working Group.

**Disclaimer** The views expressed are those of the authors and not necessarily those of the NIHR or the Department of Health and Social Care.

**Competing interests** None declared.

**Patient and public involvement** Patients and/or the public were involved in the design, or conduct, or reporting, or dissemination plans of this research. Refer to the Methods section for further details.

**Patient consent for publication** Not applicable.

**Provenance and peer review** Not commissioned; externally peer reviewed.

**ORCID iDs**
Emily Owen-Boukra http://orcid.org/0000-0001-5558-8567
Bryan Burford http://orcid.org/0000-0003-4687-7556
Claire Duddy http://orcid.org/0000-0002-7083-6589
Claire Goodman http://orcid.org/0000-0002-8938-4893
Tim Rapley http://orcid.org/0000-0003-4836-4279

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
