## [Reviewer comments · BMJ Open]

ARTICLE DETAILS

TITLE (PROVISIONAL)	General practitioner workforce sustainability to maximise effective and equitable patient care: a realist review protocol.
AUTHORS	Park, Sophie; Owen, Emily; Burford, Bryan; Cohen, Tanya; Duddy, Claire; Dunn, Harry; Fadia, Vacha; Goodman, Claire; Henry, Cecily; Lamb, Elizabeth; Ogden, Margaret; Rapley, Tim; Rees, Eliot; Vance, Gillian; Wong, Geoff

VERSION 1 – REVIEW

REVIEWER	McDonald, Terrence University of Calgary, Department of Family Medicine
REVIEW RETURNED	26-Jun-2023

GENERAL COMMENTS	Thank for for the opportunity to review this important (protocol) study. As a Canadian GP and health services researcher this topic resonates very well and I look forward to the outcomes of your proposed analysis. A few comments for your review and to hopefully strengthen your work, particularly for a broader international audience that will undoubtedly benefit from this work. I would suggest some definitions on both equitable access equitable and effective(patient care), some context and focal definition will be very helpful to situated the rationale and importance of this work. Further, providing context to the NHS and how GPs (and the teams they work with) are placed will help provide an understanding to the work and in particular, the three proposed case studies. Finally two items for your consideration on context that I believe will again strengthen this work. The first is to consider providing a brief overview of the current GP demographics (not just the change in count, but urban and rural disparities). Are issues faced in the UK, like other similar jurisdictions in the western world where many GPs are aged and nearing retirement? This will help add context to the 'why' portion of this work. The second, are both the stakeholder and audience this work targets, although outlined, it remained unclear to me who might be
--

	listening and able to change policy, perhaps some historical context again on GP engagement and policy drivers will rally reinforce the outcome of this proposed framework. Regarding the proposed method framework selected, I might suggest a very brief rationale for the selection and pros/cons around using it for this particular set of questions. Best of luck on your work.
--	--

REVIEWER	Burgers, JS Nederlands Huisartsen Genootschap
REVIEW RETURNED	11-Aug-2023

GENERAL COMMENTS	This well-written manuscript describes a protocol of a realist review exploring the healthcare system aspects that affect GP workforce sustainability. It is very relevant in the context of the crisis in health care due to increasing shortage of GPs as the authors describe. This is not only a problem in UK but also in other countries in and outside Europe. The authors could describe the challenge to keep the workforce in primary care in other countries as well to increase the interest of the paper for readers outside UK. In addition, I have two other general comments:  1. The shortage of GPs is not unique. The shortage of nurses and other professionals in health care is alarming as well. In other domains of society, e.g. education, policy and security, there are also shortages in workforce due to aging, more part-time work, and other factors. The authors should specify which aspects are unique for GPs compared to other healthcare workers in primary and secondary care. For instance, is it more attractive to work in hospital and why? The problems in other domains of society could be acknowledged in the Discussion. 2. The only limitation mentioned is the language. The authors could describe more in detail the limitations of the realist review approach compared to other types of review. Consider to include a separate paragraph 'Strength and limitations' in the Discussion. I have a few more specific comments:  - p.5,r.74: please explain the suggestion that 'resilience or failure' influence GPs decision to stay or leave. This might be opinion-based. - p.6,r.86: is reference 86 correct? It does not follow numerically on ref 9. - p.176-180: this is very theoretical and quite vague. An example could help making it more concrete. Do the authors have specific hypotheses? - p.10,r.190: consider to include nurses and medical specialists that closely collaborate with primary care teams. - p.10, r.196: person-centred care is a core value of primary care. How does this relates to the movement of personalised medicine adopted in secondary care? What is unique for GP care? - p.11, r.208: this case is recognisable for GPs in other countries as well. - p.12, r.227: which factors could explain such differences between these areas? - p.14, r.270: the word cloud does not contain words in different size and is difficult to read due to light colours. Consider to skip this figure and explain more in the text.
--

	- p.14, r.273: figure 3 largely overlaps with figure 1. I would suggest one figure with more explanation in the legend of the figure or in text. - p.14, r.282: only additional databases are mentioned. What are the main databases searched? - p.14, r.283: please provide examples of free text and subject headings. Preferably, a design of a reproducible search strategy is included. - p.15, r.290: who are 'personal contacts'? Particular networks? How do you deal with potential conflict of interests?
--	---

REVIEWER	Pereira Gray, Denis St Leonard's Research Practice
REVIEW RETURNED	12-Aug-2023

GENERAL COMMENTS	Thank you for inviting me to review this submission which I am pleased to do. NO CONFLICT OF INTEREST I know the first author but have never researched with her and have no shared grant or financial interest with her or any of the author team. Although cited by them, I believe I am free of any conflict of interest and am free to write a review. STRENGTHS This submission has many strengths: The topic is particularly important given the crisis in general practice with falling numbers of GPs, particularly the retention problem in the NHS, and the importance of the GP role. Their focus on sustainability is correct as recruitment is not enough and retention is very important. A realist review is logical given the complexity of the topic. The backing of the School of Primary Care is an advantage. Taking relationship based professional satisfaction into account may be important. Patient participation is strong. Including Kajira -Montag et al (I declare a collaboration) is impressive as that work has not yet been published in the medical literature. They are right to do so as this work from Cambridge shows that continuity of GP care is associated with a reduction of overall practice workload. This is potentially important for sustainability in general practice. LIMITATIONS Whilst good patient participation is a strength, the essence of the research question is GP sustainability. Since loss of retention follows private feelings in GPs' minds, GP participation in the main categories of GP needs to be involved and their participation could be better described. The Kajira-Montag et al reference is incomplete. This study faces a problem. Negative GP feelings- leading to dissatisfaction and then thoughts of leaving practice or retiring early have attracted attention and publications, which they cite. However, GPs with generally good relationships with their patients, strong continuity of GP care, and relatively high professional satisfaction have been much less studied. Yet they are the group which may offer a model for GP sustainability. The authors may need to give more thought to how they can, in addition to their cited references, find and include the views of GPs like, Dr Anna Graham of the Horfield Practice, Bristol who often states "I love my job". RECOMMENDATION
---

	This is a particularly important and timely research question and in principle this proposal should be supported. The submission could be further strengthened if the authors could review the points made above.
--	---

REVIEWER	Morishita, Mariko Kyoto University, Medical Education Center
REVIEW RETURNED	03-Sep-2023

GENERAL COMMENTS	Thank you for allowing me to review the protocol paper. The article is a protocol of a realist review for developing a programme theory that clarifies relationships among aspects of the health care system and GP workforce sustainability. The realist review has enough power to visualise contexts, mechanisms and outcomes surrounding GP workforce. The authors developed the first programme theories in this protocol through the literature review and conversations with PPI and stakeholders. That gave the reviewer a new understanding of how GPs work (or quit working) and how systems could affect their work and life. Especially case studies are interesting and depict well how GPs struggle with their work. The reviewer believes the study based on the protocol will contribute to discussions and policy makings around GP workforce sustainability. However, the reviewer finds that this protocol paper can be improved more before its publication. The realist review method seems to be applied to develop the protocol. If so, the reviewer would like to recommend the authors add more description about the method to create the first programme theories in this protocol. In addition, the description about the method can inform readers of the rationales of a process to develop the first programme theories. The attached document contains the reviewer's comments that may be addressed before resubmission. 1) Introduction Reading the Introduction, the reviewer understood how GPs' works are essential and beneficial for patients and systems. The pile of evidence showing how GPs work have been accumulated. Still, that information was analysed in fragmented ways, about the number of GPs, the level of emotional distress, and lengths of time for consultations, for example. As the authors noted, a system approach could delineate and examine the milieu of the GP workforce. The reviewer presumes that a programme theory is one of the concepts included in a system approach. As the reviewer is not in a discipline related to policy-making, programme theory is not a familiar term. This protocol will be read by scholars in policy-making, as well as GPs and academics in other areas. What the term programme theory means can be explained in the Introduction, which could help readers understand what the authors mean when they write the term "a system approach." 2) Methods The reviewer thinks that the order of words, review aim, questions and objective should be
--

	changed to align with the order of each section, aim, research objectives and research questions. 3) Approach The approach can be written more clearly to let readers know how the authors developed the programme theories drawn in Fig. 2 and Fig. 3. The reviewer recommends that the authors add a description of the strategy or method they adopted when they developed the initial and expanded programme theory. The reviewer presumes that a realist review may be applied to create the protocol. If so, that needs to be clearly mentioned. There seem to be several steps, including initial scoping literature searches, discussions with PPI and stakeholders, producing cases and asking for GPs' joy in their works. If the authors could show a whole picture of what they did to create the program theories in this protocol, readers could follow the logic in this section more easily. 4)Patient and Public Involvement p. 10, line 193- The authors described what their PPI co-applicants and stakeholders prioritised, which was interesting and unique. The cases delineated well how GPs work and their working milieu. It seems complicated to depict how the authors reflected that information in the initial programme theory. The reviewer felt difficulty in understanding relationships among the literature searched in the previous section, the discussions and cases by PPI and stakeholders, and the programme theories. As mentioned above, writing the method more clearly could help readers understand the logic behind the process of creating programme theories. p.13, line 255- The authors explored the joy of GP's work, which is one of the interesting points in this protocol. The reviewer presumes that their joy can be a decisive factor for GPs to continue to work. It is understandable. However, rationales for choosing this theme, joy of GPs' work may be explained. The reviewer wonders why the authors did not choose other themes, such as reasons for GPs to continue to work or factors they dislike in their works.
--	---

VERSION 1 – AUTHOR RESPONSE

Reviewer 1 1. Thank for the opportunity to review this important (protocol) study. As a Canadian GP and health services researcher, this topic resonates	Thank you for your insightful feedback and detailed review.	
---	--	--

very well and I look forward to the outcomes of your proposed analysis. A few comments for your review and to hopefully strengthen your work, particularly for a broader international audience that will undoubtedly benefit from this work.		
1. I would suggest some definitions on both equitable access equitable and effective (patient care), some context and focal definition will be very helpful to situate the rationale and importance of this work.	We have provided definitions of equitable and effective patient care.	L113-119.
2. Further, providing context to the NHS and how GPs (and the teams they work with) are placed will help provide an understanding to the work and in particular, the three proposed case studies.	We have provided more context here. Importantly, however, we are mindful that the organisation and delivery of care may vary depending on local population needs and available resources. Therefore, 'good' may vary across different settings and contexts. We have chosen a realist approach as this allows us to examine the complex and dynamic interplay between contextual factors, mechanisms, and outcomes.	L70-74.
3. Finally, two items for your consideration on context that I believe will again strengthen this work. The first is to consider providing a brief overview of the current GP demographics (not just the change in count, but urban and rural disparities).	We have added information and provided references to highlight the challenges of attracting and retaining GPs in rural and coastal areas.	L89-90.
4. Are issues faced in the UK, like other similar jurisdictions in the western world where many GPs are aged and nearing retirement? This will help add context to the 'why' portion of this work. The second, are both the stakeholder and audience this work targets, although outlined, it remained unclear to me who might be listening and able to change policy, perhaps some historical context again on GP engagement and policy drivers will really reinforce the outcome of this proposed framework.	Yes, thank you. We have added a sentence to emphasise that many GPs are nearing the traditional retirement age. Re: the stakeholders and audience our review targets – we aim to target policymakers, health system decision-makers, and commissioners. We have added more information to emphasise that there are on-going national and international debates about what general practice of the future should look like. We hope the findings from our review can influence political discussions, contribute to the scholarship of the field, and to a	L88. L388. L399-403.

	more sustainable general practice future.	
5. Regarding the proposed method framework selected, I might suggest a very brief rationale for the selection and pros/cons around using it for this particular set of questions.	Thank you – we have highlighted the strengths of the realist approach within our introduction section. We have added ‘realist’ to make this clearer (i.e., L108-113, “By taking a system and theory-driven realist approach to understanding what works, in which contextual circumstances, and to what extent, we aim to make visible the often unsaid or implicit issues that influence working environments, relationships, and organisational culture. The findings from our review will offer structural and organisational recommendations to inform sustainable, context-specific ways of future working, which can underpin effective and equitable patient care.” We have also outlined a limitation of our realist approach within the ‘strengths and limitations’ section. “Our context-mechanism-outcome configurations and refined programme theory may be limited by the availability, quality, and richness of existing identified literature in this field.”	L108. L54-56.
Reviewer 2: 1. This well-written manuscript describes a protocol of a realist review exploring the healthcare system aspects that affect GP workforce sustainability. It is very relevant in the context of the crisis in health care due to increasing shortage of GPs as the authors describe. This is not only a problem in UK but also in other countries in and outside Europe. The authors could describe the challenge to keep the workforce in primary care in other countries as well to increase the interest of the paper for readers outside UK.	Thank you very much for your feedback and detailed review. While this problem is not exclusive to the UK, there may be some factors which are unique to the UK setting, and other factors which are relevant and transferable elsewhere. Importantly, a realist approach enables us to explore the relationship between contexts, mechanisms, and outcomes, developing causal explanations and generating new insights. These insights can then inform work elsewhere, as well as in the UK. Within our introduction section, we have added the following sentence “the challenges of recruiting and retaining GPs in rural and coastal areas is well recognised, both nationally and	L89-91.

	internationally.” We hope this will increase the interest of the protocol paper for readers outside the UK setting.	
2. In addition, I have two other general comments: 1. The shortage of GPs is not unique. The shortage of nurses and other professionals in health care is alarming as well. In other domains of society, e.g. education, policy and security, there are also shortages in workforce due to aging, more part-time work, and other factors. The authors should specify which aspects are unique for GPs compared to other healthcare workers in primary and secondary care. For instance, is it more attractive to work in hospital and why? The problems in other domains of society could be acknowledged in the Discussion. 2. The only limitation mentioned is the language. The authors could describe more in detail the limitations of the realist review approach compared to other types of review. Consider to include a separate paragraph 'Strength and limitations' in the Discussion.	Because we are using a systems-level approach to look at this problem, we are likely to identify some issues which are not specific (but are relevant) to general practitioners. However, we will constantly critically examine what the available literature can and cannot tell us (i.e. knowledge, gaps, suggestions for future research). Thanks for your feedback. : We have added an additional limitation... “Our context-mechanism-outcome configurations and refined programme theory may be limited by the availability, quality, and richness of existing literature in this field.”	L54-56.
3. p.5,r.74: please explain the suggestion that 'resilience or failure' influence GPs decision to stay or leave. This might be opinion-based.	When GPs decide to either stay or leave the profession, it is often perceived as an individual choice that depends on their resilience and ability to persevere in difficult situations. Yet, this does not consider (or blame) the structural and system level factors that may influence and / or force GPs to leave. Our review seeks to move the focus beyond whether the individual has survived employment in a system, to look at the relationship and interactions between individuals and the environment / culture in which they work.	
4. p.6,r.86: is reference 86 correct? It does not follow numerically on ref 9.	Thank you for spotting this, we have changed so it now follows numerically.	L94.

5. p.176-180: this is very theoretical and quite vague. An example could help making it more concrete. Do the authors have specific hypotheses?	A realist approach begins by developing an initial programme theory (this describes our assumptions about important influences on GP workforce sustainability). We have included an example of our initial programme theory in Figure 1 (for instance, we have theorised how organisational size and management, shifting role mix, continuity with colleagues and patients may impact GP workforce sustainability). Throughout our review, we will modify and refine our initial programme theory based on relevant literature (qual, quant, mixed-methods evidence, and grey literature), and regular PPI / stakeholder engagement. We will provide causal explanations by developing context-mechanism-outcome configurations. We are therefore responding to the available data in a configurative way: looking across and forming connections between identified data (explanatory claims – what is it about X that leads to Y?).	
6. p.10,r.190: consider to include nurses and medical specialists that closely collaborate with primary care teams.	This is helpful – thank you. Within our core research team, we do have a district nurse (by background) and a consultant paediatrician. Our stakeholder group will also include recruitment of key individuals outside of general practice (e.g. paediatricians, professional services, physician associates etc.).	
7. p.10, r.196: person-centred care is a core value of primary care. How does this relate to the movement of personalised medicine adopted in secondary care? What is unique for GP care?	The traditions of general practice in the UK have historically prioritised holistic and person-centred care (i.e. able to support patients with symptoms that do not fit within a disease label, as well as people navigating multiple, co-existing, or even competing disease management plans). We have added a sentence to include 'person-centred' within our introduction section, to highlight that general practice provides comprehensive (a patient may present with any type of problem), person-centred	L70-74.

	(holistic caring and individualised care), and universal (everyone can attend) access to healthcare.	
8. p.11, r.208: this case is recognisable for GPs in other countries as well.	Yes. We have re-worded this sentence.	L220-223.
9. p.12, r.227: which factors could explain such differences between these areas?	A realist review enables us to look at the connections between context, mechanisms, and outcomes. Through analysing our identified data, we can then look for explanatory patterns, differences, and commonalities which may help to answer this question.	
10. p.14, r.270: the word cloud does not contain words in different size and is difficult to read due to light colours. Consider to skip this figure and explain more in the text.	Thanks, this is helpful. We have edited the word cloud visualisation (Figure 2) to improve readability.	Figure 2.
11. p.14, r.273: figure 3 largely overlaps with figure 1. I would suggest one figure with more explanation in the legend of the figure or in text.	We have made it clearer what has been added (i.e. new changes in Figure 3 are now in orange). The additions in the 'Recent Changes' box have also been underlined in addition to the new colour.	Figure 1 and 3.
12. p.14, r.282: only additional databases are mentioned. What are the main databases searched?	The main databases we will search include: MEDLINE; Embase; PsychINFO; CINAHL; HMIC; and Web of Science. We have added this information.	L292-298.
13. p.14, r.283: please provide examples of free text and subject headings. Preferably, a design of a reproducible search strategy is included.	Thanks. We have provided examples of free text and subject headings. We have also included a design of a reproducible search strategy in Supplementary File 1.	L296-298. Supplementary File 1.
14. p.15, r.290: who are 'personal contacts'? Particular networks? How do you deal with potential conflict of interests?	For clarity, we have changed 'personal contacts' to 'our PPI and stakeholder members'. In realist reviews, data do need to be interpreted, but to reduce any potential threats to the validity of our findings, we have built in a range of processes. These include: actively looking for disconfirming data; discussion within the project team and; sharing our findings with stakeholders and PPI.	L304-305.
Reviewer 3		
This submission has many strengths: The topic is particularly important given the crisis in general practice with falling numbers of GPs, particularly the retention problem in the	Thank you very much for your feedback.	

NHS, and the importance of the GP role. Their focus on sustainability is correct as recruitment is not enough and retention is very important. A realist review is logical given the complexity of the topic. The backing of the School of Primary Care is an advantage. Taking relationship based professional satisfaction into account may be important. Patient participation is strong. Including Kajira -Montag et al (I declare a collaboration) is impressive as that work has not yet been published in the medical literature. They are right to do so as this work from Cambridge shows that continuity of GP care is associated with a reduction of overall practice workload. This is potentially important for sustainability in general practice.		
Limitations Whilst good patient participation is a strength, the essence of the research question is GP sustainability. Since loss of retention follows private feelings in GPs' minds, GP participation in the main categories of GP needs to be involved and their participation could be better described. The Kajira-Montag et al reference is incomplete. This study faces a problem. Negative GP feelings- leading to dissatisfaction and then thoughts of leaving practice or retiring early have attracted attention and publications, which they cite. However, GPs with generally good relationships with their patients, strong continuity of GP care, and relatively high professional satisfaction have been much less studied. Yet they are the group which may offer a model for GP sustainability. The authors may need to give more thought to how they can, in addition to their cited references, find and include the views of GPs like, Dr Anna	Thank you. We have GPs within our core research team, and we have also recruited GPs within our stakeholder group in addition to paediatricians, professional services, physician associates etc. Thanks for pointing this out - We have referenced the following working paper by Harshita Kajaria-Montag et al (reference 18). - http://ssrn.com/abstract=3868231 We agree with this comment. We have, therefore, framed our initial stakeholder engagement work around 'joy in general practice' to ensure a balance between problems and retention factors. Thank you. We have referenced written evidence by Dr Anna Graham stating how she loves her job and emphasising the importance of personalised, patient-centred care (reference 36).	L208.

Graham of the Horfield Practice, Bristol who often states, "I love my job". RECOMMENDATION This is a particularly important and timely research question and in principle this proposal should be supported. The submission could be further strengthened if the authors could review the points made above.		
Reviewer 4		
Thank you for allowing me to review the protocol paper. The article is a protocol of a realist review for developing a programme theory that clarifies relationships among aspects of the health care system and GP workforce sustainability. The realist review has enough power to visualise contexts, mechanisms and outcomes surrounding GP workforce. The authors developed the first programme theories in this protocol through the literature review and conversations with PPI and stakeholders. That gave the reviewer a new understanding of how GPs work (or quit working) and how systems could affect their work and life. Especially case studies are interesting and depict well how GPs struggle with their work. The reviewer believes the study based on the protocol will contribute to discussions and policy makings around GP workforce sustainability. However, the reviewer finds that this protocol paper can be improved more before its publication. The realist review method seems to be applied to develop the protocol. If so, the reviewer would like to recommend the authors add more description about the method to create the first programme theories in this protocol. In addition, the description about the method can inform readers of the rationales of a process to develop the first programme theories. The attached document contains the	Thank you very much for your detailed review and insightful feedback. The initial programme theory was developed through initial engagement and immersion with the literature (and preliminary consultation with stakeholders and patients) and formed the basis of our funding application. We then carried out more detailed PPI and stakeholder work to expand the initial programme theory, which became the starting point for our review.	L184-191.

reviewer's comments that may be addressed before resubmission.		
Introduction Reading the Introduction, the reviewer understood how GPs' works are essential and beneficial for patients and systems. The pile of evidence showing how GPs work have been accumulated. Still, that information was analysed in fragmented ways, about the number of GPs, the level of emotional distress, and lengths of time for consultations, for example. As the authors noted, a system approach could delineate and examine the milieu of the GP workforce. The reviewer presumes that a programme theory is one of the concepts included in a system approach. As the reviewer is not in a discipline related to policy-making, programme theory is not a familiar term. This protocol will be read by scholars in policy-making, as well as GPs and academics in other areas. What the term programme theory means can be explained in the Introduction, which could help readers understand what the authors mean when they write the term "a system approach."	Thanks for your feedback. In realist reviews, part of the process includes a methodological step of being critically reflective about assumptions early in the process, so the journey and building blocks / steps of the review are clear to readers. This takes the form of an initial programme theory – which outlines our assumptions about important influences on GP workforce sustainability. Throughout the review, we will modify and refine our initial programme theory based on relevant literature (qual, quant, mixed-methods evidence, and grey literature), and regular PPI / stakeholder engagement. We will provide causal explanations by developing context-mechanism-outcome configurations and an evidenced-based refined programme theory.	L254-255. L364-366.
Methods The reviewer thinks that the order of words, review aim, questions and objective should be changed to align with the order of each section, aim, research objectives and research questions.	This is helpful, thank you, we have corrected.	L121.
Approach The approach can be written more clearly to let readers know how the authors developed the programme theories drawn in Fig. 2 and Fig. 3. The reviewer recommends that the authors add a description of the strategy or method they adopted when they developed the initial and expanded programme theory. The reviewer presumes that a	We hope our answer above (to the introduction question) clarifies this.	

realist review may be applied to create the protocol. If so, that needs to be clearly mentioned. There seem to be several steps, including initial scoping literature searches, discussions with PPI and stakeholders, producing cases and asking for GPs' joy in their works. If the authors could show a whole picture of what they did to create the program theories in this protocol, readers could follow the logic in this section more easily.		
Patient and Public Involvement p. 10, line 193- The authors described what their PPI co-applicants and stakeholders prioritised, which was interesting and unique. The cases delineated well how GPs work and their working milieu. It seems complicated to depict how the authors reflected that information in the initial programme theory. The reviewer felt difficulty in understanding relationships among the literature searched in the previous section, the discussions and cases by PPI and stakeholders, and the programme theories. As mentioned above, writing the method more clearly could help readers understand the logic behind the process of creating programme theories.	The processes we are using in this review follows that set out by Ray Pawson and which are commonly used in realist reviews - Pawson R. Evidence-Based Policy: A Realist Perspective London: SAGE Publications.; 2006.	
p.13, line 255- The authors explored the joy of GP's work, which is one of the interesting points in this protocol. The reviewer presumes that their joy can be a decisive factor for GPs to continue to work. It is understandable. However, rationales for choosing this theme, joy of GPs' work may be explained. The reviewer wonders why the authors did not choose other themes, such as reasons for GPs to continue to work or factors they dislike in their works.	The initial literature we identified was predominantly negative, focusing on reasons why GPs leave the profession. We recognise that many GPs continue to work (and even enjoy) their work. As such, we wanted to reflect these 'pull' factors in our review, in addition to the 'push' factors. As we are looking at the system as a whole, it is important to look at the interactions and connections experienced by GPs, as well as individual factors (e.g. resilience, burnout). The concept of joy is one of a number of stakeholder inputs into the initial programme theory. Each will be explored in	

	relation to existing and identified evidence.	
--	---	--

VERSION 2 – REVIEW

REVIEWER	McDonald, Terrence University of Calgary, Department of Family Medicine
REVIEW RETURNED	07-Feb-2024

GENERAL COMMENTS	Thank you for the opportunity to re-review this very valuable work. I have a few comments for your consideration that I hope might serve to strengthen it further. I look forward to the outcomes of this work. Kindest regards, TM. 1. From Introduction: Lines 93-94: Is this statement complete? It appears to elude to a relative change to another year or time period? "...with 37.2 million consultations estimated to have been delivered by practices and primary care networks in October 2023". A brief qualifier might be of value. 2. From the Discussion: Lines 399-401. This opening sentence is rather robust and is the first occasion it seems where the COVID-19 Pandemic and its effect are referred to. Are there references that could be applied here? Reflecting back to parts of the Introduction, which is very well written may be of value to reinforce the rationale for this work. In this regard, a very brief expansion of the Discussion will assist to reaffirm the key takeaways for readers. If changes in the GP workforce were observed from the COVID-19, which seem likely, then this might be introduced earlier and tied in. 3. Re: the term 'Continuity' within the text and figures, as you aware, there are multiple types. As a GP, it is the relational or interpersonal continuity with patients, staff and colleagues that is very impactful in regard to ones career satisfaction, longevity and wellness. I've included this reference, although an older paper, the author outlines different forms of continuity, for your consideration. Wall EM. Continuity of care and family medicine: definition, determinants, and relationship to outcome. The Journal of family practice. 1981;13(5):655-64. https://cdn.mdedge.com/files/s3fs-public/jfp-archived-issues/1981-volume_12-13/JFP_1981-10_v13_i5_continuity-of-care-and-family-medicine-.pdf
--

REVIEWER	Burgers, JS Nederlands Huisartsen Genootschap
REVIEW RETURNED	29-Jan-2024

GENERAL COMMENTS	The authors integrated the reviewers' comments well. Strengths of this study are the patient and public involvement and the potential solutions for addressing the workforce problem. The outcomes of this review is very relevant for UK and other countries. I hope that preliminary results will be presented before the end of the study in December 2025 as the crisis will go on and may be even getting
---

	worse. I only have some additional suggestions about the manuscript that can be considered:  - The terms 'context' and 'mechanism' are very broad. A definition and/or examples are helpful to make it more concrete. Consider to illustrate the conceptual framework with a figure including these terms. This could include the relationship between contextual factors, mechanisms, and reported outcomes. - p. 8, r. 139: This is difficult to read. Consider reformulation. - p. 9, r. 177: are 'circumstances' specifications of the 'context'? Explain or use this when defining 'context'.
--	--

REVIEWER	Morishita, Mariko Kyoto University, Medical Education Center
REVIEW RETURNED	19-Feb-2024

GENERAL COMMENTS	Thank you for your revised manuscript. The reviewer found that the manuscript has improved profoundly following the reviewers' comments. The responses to the comments are understandable and plausible. However, the reviewer would like to suggest some points be revised. Introduction If "programme theory" is not a general term, what "programme theory" means should be explained within a description of a realist review. From the reviewer's understanding of realism and realist review, a programme theory can explain the causality of focused phenomena (or research themes). This sort of explanation will help readers understand more clearly what the figures (1 and 3) of the programme theory could mean. How about inserting the description of "what 'programme theory' is" in the latter part of the introduction? Line 265- The reviewer asked why GP's joy was used for expanding the initial programme theory, previously. The response is plausible. (According to the response, the authors focused on positive aspects of GP's work in the expanding stage of the initial programme theory.) The reviewer thinks that this reason can be added to this section to show how the authors systematically elaborate the programme theory through discussions with stakeholders and engaging PPI.
--

VERSION 2 – AUTHOR RESPONSE

Reviewer 2		
1. The authors integrated the reviewers' comments well. Strengths of this study are the patient and public involvement and the potential solutions for addressing the workforce problem. The outcomes of this review is very relevant for UK and other countries. I hope that preliminary results will	Thank you very much for your feedback.	

be presented before the end of the study in December 2025 as the crisis will go on and may be even getting worse. I only have some additional suggestions about the manuscript that can be considered		
2. The terms 'context' and 'mechanism' are very broad. A definition and/or examples are helpful to make it more concrete. Consider to illustrate the conceptual framework with a figure including these terms. This could include the relationship between contextual factors, mechanisms, and reported outcomes.	Thank you for your feedback. We have provided succinct definitions of context, mechanisms, and outcomes. We have also provided examples of potential contexts (e.g., organisation size, employment type, practice skill-mix, and nature of patient contact) and mechanisms (e.g., continuity, peer dialogue, clinical autonomy).	L147-149. L259-263.
3. p. 8, r. 139: This is difficult to read. Consider reformulation.	Thank you. We have re-worded this. What are the important contexts which influence whether different mechanisms produce intended and/or unintended outcomes in GP workforce sustainability?	L139-140.
4. p. 9, r. 177: are 'circumstances' specifications of the 'context'? Explain or use this when defining 'context'.	Thanks. When defining the word 'context' we have stated 'conditions or circumstances'. However, for clarity, we have changed circumstances to 'contexts' within this sentence. We hope this is clearer now.	L147-148. L180.
Reviewer 1		

1. Thank you for the opportunity to re-review this very valuable work. I have a few comments for your consideration that I hope might serve to strengthen it further. I look forward to the outcomes of this work.	Thank you very much for your feedback and comments.	
2. From Introduction: Lines 93-94: Is this statement complete? It appears to elude to a relative change to another year or time period? "...with 37.2 million consultations estimated to have been delivered by practices and primary care networks in October 2023". A brief qualifier might be of value.	We have provided more information.	L91-93.
3. From the Discussion: Lines 399-401. This opening sentence is rather robust and is the first occasion it seems where the COVID-19 Pandemic and its effect are referred to. Are there references that could be applied here? Reflecting back to parts of the Introduction, which is very well written may be of value to reinforce the rationale for this work. In this regard, a very brief expansion of the Discussion will assist to reaffirm the key takeaways for readers. If changes in the GP workforce were observed from the COVID-19, which seem likely, then this might be introduced earlier and tied in.	Thank you. We have added references (45, 46, 47). We have also referred to the COVID-19 pandemic in our introduction section. We have expanded our discussion section to reaffirm the key takeaways for readers.	L409. L93. L409-412.

4. Re: the term 'Continuity' within the text and figures, as you are aware, there are multiple types. As a GP, it is the relational or interpersonal continuity with patients, staff and colleagues that is very impactful in regard to ones career satisfaction, longevity and wellness. I've included this reference, although an older paper, the author outlines different forms of continuity, for your consideration. Wall EM. Continuity of care and family medicine: definition, determinants, and relationship to outcome. The Journal of family practice. 1981;13(5):655-64. https://cdn.mdedge.com/files/s3fs-public/jfp-archived-issues/1981-volume_12-13/JFP_1981-10_v13_i5_continuity-of-care-and-family-medicine-.pdf	In the manuscript, we have made it explicit what types of continuity the referenced articles are referring to (or implying) (i.e. relational). For instance, the review by Long and colleagues (2020) describes how “GPs job satisfaction directly relates to the quality of the doctor-patient relationship, with more time available for GPs to spend with their patients (on-going personal relationships), being associated with better job satisfaction. “Lack of time with patients is perceived to compromise the ability to practise patient-centred care and undermines GPs professional autonomy and values, resulting in further diminished job satisfaction.” The review by Andah and colleagues (2021) also describes decreased time with patients and a loss of continuity contributing to low morale. We have also provided examples of the	L166-167.

	different types of continuity (e.g., relational, informational, longitudinal, and management types of continuity) and referenced the valuable paper by Eric Wall in addition to a multidisciplinary review by Haggerty et al. (2003). Thank you very much for sharing the paper by Wall. Re: the figures – these are our ‘initial’ or ‘emerging’ programme theories. Once we have analysed our data and completed our review, we will produce a finalised and refined programme theory. This will make explicit the different types of continuity, which may be the most important to keep, sustain, and enable GPs to flourish within their work.	L261-262.
Reviewer 4 1. Thank you for your revised manuscript. The reviewer found that the manuscript has improved profoundly following the reviewers' comments. The responses to the comments are understandable and plausible. However, the reviewer would like to suggest some points be revised.	Thank you very much for your feedback.	
2. Introduction If "programme theory" is not a general term, what "programme theory" means should be explained within a description of a realist review. From the reviewer's understanding of realism and realist review, a programme theory can explain the causality of focused phenomena (or research themes). This sort of explanation will help readers understand more clearly what the figures (1 and 3) of the programme	Thank you. We have provided a definition of a ‘programme theory’. A programme theory explains how a programme, intervention, or process works (e.g. how we can develop a more	L145-147.

theory could mean. How about inserting the description of "what 'programme theory' is" in the latter part of the introduction?	sustainable GP workforce). We have also provided a reference (Wong et al., 2013) for further information. We hope this is clearer.	
3. Line 265- The reviewer asked why GP's joy was used for expanding the initial programme theory, previously. The response is plausible. (According to the response, the authors focused on positive aspects of GP's work in the expanding stage of the initial programme theory.) The reviewer thinks that this reason can be added to this section to show how the authors systematically elaborate the programme theory through discussions with stakeholders and engaging PPI.	Thank you. We have added this information. "We focused on joy (and meaning making) as most of the literature we identified in our initial scoping searches was predominantly negative, highlighting the reasons why GPs leave or consider leaving direct patient care. We recognise that many GPs continue to enjoy and derive pleasure from their work (43). As such, our aim throughout the PPI and stakeholder discussions was to understand the factors that keep GPs engaged meaningfully in their work."	L273-278.

VERSION 3 – REVIEW

REVIEWER	McDonald, Terrence University of Calgary, Department of Family Medicine
REVIEW RETURNED	11-Mar-2024
GENERAL COMMENTS	Thank you again for the opportunity to review this valuable work. I look forward to the results of this project.
REVIEWER	Morishita, Mariko Kyoto University, Medical Education Center
REVIEW RETURNED	31-Mar-2024

GENERAL COMMENTS	Thank you for allowing me to re-review the manuscript. I found that the authors revised the manuscripts well, following the reviewers' suggestions. I believe that this manuscript will be valuable and meaningful as a realist review protocol for the authors' work on exploring aspects related to GP workforce sustainability.
--

VERSION 3 – AUTHOR RESPONSE